# Associating Land Cover Changes with Patterns of Incidences of Climate-Sensitive Infections: An Example on Tick-Borne Diseases in the Nordic Area

**DOI:** 10.3390/ijerph182010963

**Published:** 2021-10-19

**Authors:** Didier G. Leibovici, Helena Bylund, Christer Björkman, Nikolay Tokarevich, Tomas Thierfelder, Birgitta Evengård, Shaun Quegan

**Affiliations:** 1School of Mathematics and Statistics, University of Sheffield, Sheffield S10 2TN, UK; s.quegan@sheffield.ac.uk; 2GeotRYcs Cie, 34000 Montpellier, France; 3Department of Ecology, Swedish University of Agricultural Sciences, 75007 Uppsala, Sweden; christer.bjorkman@slu.se; 4Laboratory of Zoonoses, St. Petersburg Pasteur Institute, 197101 St. Petersburg, Russia; zoonoses@mail.ru; 5Department of Energy and Technology, Swedish University of Agricultural Sciences, 75007 Uppsala, Sweden; tomas.thierfelder@slu.se; 6Department of Clinical Microbiology, Umeå University, 90187 Umeå, Sweden; birgitta.evengard@umu.se

**Keywords:** climate-sensitive infection, vector-borne disease, tick-borne disease, climate change, land cover, vegetation type, Nordic, Fennoscandia

## Abstract

Some of the climate-sensitive infections (CSIs) affecting humans are zoonotic vector-borne diseases, such as Lyme borreliosis (BOR) and tick-borne encephalitis (TBE), mostly linked to various species of ticks as vectors. Due to climate change, the geographical distribution of tick species, their hosts, and the prevalence of pathogens are likely to change. A recent increase in human incidences of these CSIs in the Nordic regions might indicate an expansion of the range of ticks and hosts, with vegetation changes acting as potential predictors linked to habitat suitability. In this paper, we study districts in Fennoscandia and Russia where incidences of BOR and TBE have steadily increased over the 1995–2015 period (defined as ’Well Increasing districts’). This selection is taken as a proxy for increasing the prevalence of tick-borne pathogens due to increased habitat suitability for ticks and hosts, thus simplifying the multiple factors that explain incidence variations. This approach allows vegetation types and strengths of correlation specific to the WI districts to be differentiated and compared with associations found over all districts. Land cover types and their changes found to be associated with increasing human disease incidence are described, indicating zones with potential future higher risk of these diseases. Combining vegetation cover and climate variables in regression models shows the interplay of biotic and abiotic factors linked to CSI incidences and identifies some differences between BOR and TBE. Regression model projections up until 2070 under different climate scenarios depict possible CSI progressions within the studied area and are consistent with the observed changes over the past 20 years.

## 1. Introduction

Global warming has already had a rapid and large impact on weather patterns and the local climate, especially in arctic and subarctic areas [1]. Increasing temperatures and changed patterns of precipitation will, in combination with changes in human activities and land use, have large-scale effects on terrestrial systems globally [2]. All biota are affected by local environmental conditions, and a changing climate will affect species distributions at different rates and to different degrees depending on species life history characteristics and requirements, e.g., dispersal ability and physiological tolerance to abiotic variability [3]. This will have consequences for ecosystem diversity and species composition and will affect species interactions at all trophic levels [4,5,6,7].

In the Arctic, the warming climate has already caused shifts in the range of vegetation, which could have consequences for wildlife and ecosystem services and could possibly drive feedback on climate [8]. Among multiple species that are expected and reported to shift their range of distribution are wildlife hosts of zoonotic diseases. They show an increasing share of species richness in areas heavily influenced by human activities at a global scale [9], which has implications for human health risks [10].

One of several concerns is the observed patterns of increased incidence and extended prevalence of multiple zoonotic diseases that have been observed in new areas [3]. Several of these diseases are transmitted to humans and other vertebrates by blood-feeding invertebrate vectors, e.g., mosquitoes and ticks [11]. In Fennoscandia and western Eurasia, there has been a clear extension into new areas of the distribution range of large mammals that carry ticks capable of transmitting infectious diseases to humans [12,13,14,15,16,17]. These changes in distribution range are associated with an increase in northern areas of several vector-borne zoonotic diseases [18,19].

This paper studies spatiotemporal changes in human incidence of two diseases that are mainly tick-borne: Lyme borreliosis, the bacterial disease caused by *Borrelia burgedorfi sensu lato* (BOR), and the viral disease tick-borne encephalitis (TBE). Combined with climatic factors, it investigates whether changes in land cover and vegetation type could explain the distributions of disease prevalence by linking them to the extended range of distribution of ticks and their hosts [15]. Ticks are also known to be the vector of other zoonotic diseases, such as anaplasmosis, babesiosis, and tularemia, though with different levels of importance in the transmission, e.g., babesiosis is exclusively transmitted by ticks, not tularemia. As these diseases are also transmitted in other ways, we will only briefly discuss tularemia. The most prominent tick species in the western part of the area of interest is the sheep tick *Ixodes ricinus*, while in the eastern part of the study area, the taiga tick *Ixodes persulcatus* is the dominant tick species reported in disease transmission to humans. The taiga tick has been observed to have spread in recent years and has become established in areas further west into Finland and in Sweden. Other hard-bodied tick species with proven vector competence for zoonotic diseases are starting to occur in new areas, such as ticks of the *genera Dermacentor* [20] or *Hyalomma* [21]. The prevalence and infection rate of tick-borne disease agents in ticks depends on the presence of competent vertebrate hosts that maintain the bacterial or viral agents of the disease [22].

The last IPCC special report on climate change and land (IPCC 2019, chapter 2) summarised the consequences of global warming: *“In boreal regions, for example, where projected climate change will migrate the treeline northward, increase the growing season length and thaw permafrost, regional winter warming will be enhanced by decreased surface albedo and snow, whereas warming will be dampened during the growing season due to larger evapotranspiration (high confidence).”* [23]. Greening and shrubification of the arctic have been reported in the literature as climatic impacts over the last 30 years, though with complex interpretations and interactions, for example with reindeer husbandry, climate feedback, and browning phenomena [24,25,26,27,28,29,30,31,32]. The general hypothesis of this paper is that land conditions, including vegetation types and cover, within recent and foreseen climatic changes constitute a set of potential drivers of change in disease prevalence. This hypothesis implies impacts of these drivers on the changes in vector distributions, on host distributions, and potentially on changes in human behaviour, leading to changes in the risks of being infected from ticks.

The changing climate directly affects the biodiversity and composition of ecosystems in marginal areas and allows some species to extend their distribution range while others retract or go extinct. The temperature and patterns of precipitation already changed most rapidly in northern areas and affected the distribution of plants and animals in subarctic areas [33], e.g., the observed shrubification, where willows and other shrubs grow at higher elevations above the treeline and extend into heathlands [4,34,35].

## 2. Materials and Methods

The Nordic Centre of Excellence (NCoE) CLINF [36] investigates the effects of climate change on the prevalence of infectious diseases in humans and animals in northern regions. The project has gathered information on the prevalence and incidence of potential CSIs (Climate-Sensitive Infectious diseases) in humans and animals in the north, with the aim of characterising the environmental envelopes under which disease vectors can thrive and propagate. The datasets from CLINF used in this paper are summarised in Appendix A.

The presence and spatiotemporal variations of vector-borne pathogens and the vector and host distributions are not directly addressed here. The focus is on the distribution of vegetation types coupled with climate change as providing potential drivers of changes in the incidence variations in two tick-borne diseases, Lyme borreliosis and tick-borne encephalitis. In order to establish simple correlations between the types of vegetation and disease incidence, the analysis focuses on districts where a steady increase in incidence has been observed. This selection is used as a proxy for the presence of pathogens due to the potential increased habitat suitability for the vectors and hosts, simplifying the multiple factors involved in disease incidence variations such as climate and land conditions, and nearby human activities. Ecological explanations of these links can then help in developing mechanisms to monitor these zones of increasing risk.

### 2.1. Data Analysis Aspects

This paper uses data for the time period 1985–2016 for which the CLINF project gathered yearly incidence rates of BOR and TBE in national districts in Norway, Sweden, Finland, and districts in western Russia. Corresponding land cover data and climate variables from global atmospheric reanalysis (ESA-CCI land cover and ERA-interim data) with appropriate transformations, e.g., maximum summer temperature and land cover conversion to plant functional type (PFT) (Appendix A), are used as explanatory variables.

For each disease, each district time series of incidences was smoothed using a non-parametric local regression (loess method [37]) and then visually assigned to one of the five classes of simple patterns: well-increasing (WI), increase–plateau (IP), increase–decrease (ID), plateau (P), and decrease–plateau (DP). A range of statistical methods is applied to give insights into the associations between variables, to decompose variations, and to model log-linear regressions, e.g., simple boxplots, surface interpolations, multiway correspondence analysis, and Gaussian and negative binomial regression models.

Land cover changes over time and differences between districts were first assessed using multiway methods [38,39]. The generic multiway method decomposes the variability in a multi-table dataset (i.e., one represented as a *k*-dimensional array) in an approach similar to Principal Component Analysis (PCA) of a two-dimensional data table. This allows, for example, for an analysis of the major contributors to the variation in land cover changes over time for the districts studied and produces a spatial map, a time series, and a vegetation weighting component for each extracted fraction of total variability. Highlighted land covers are subsequently described using boxplots and interpolation diagrams. Projections of incidences to 2070 are obtained from the selected regression models trained using the historical data, along with projected land surface simulations from the ORCHIDEE model [40] and climate indicators under the RCP4.5 and RCP8.5 scenarios (Appendix B).

### 2.2. Ecological Aspects of Tick-Borne Diseases

This paper investigates if districts in northern Europe and western Russia where the incidence of tick-borne encephalitis (TBE) and Lyme’s disease (BOR) in humans have increased markedly during recent decades have also undergone changes in land cover (vegetation) compared with districts where the incidence of these diseases shows other types of patterns.

The presence of vertebrate hosts providing blood meals is essential in the life cycle of *Ixodes spp.* ticks. Tick larvae and nymphs mainly feed on small mammals and ground-feeding birds, while the adult ticks prefer to take their blood meal from larger mammals such as ungulates [22]. In northern Europe, the entire life cycle usually takes 2–3 years but can take longer; see [41] for more details on life cycle and hosts. The distribution of ticks and viable tick populations is therefore linked to habitats with climatic conditions suitable for ticks where their hosts reside when foraging [16,18,42,43]. A common description of typical tick habitat is a mixture of forest, bush vegetation, and open grassland [14,44]. The forest in such landscapes could be deciduous, coniferous, or a mixture of these, which provides foraging and shelter habitats for tick vertebrate hosts and suitable microclimatic conditions for the ticks on the forest floor. In these areas, all life stages of the *Ixodes* ticks have the possibility to find a host for their blood meals, from ground feeding birds and small mammals preferred as hosts by the larval and nymph stage to deer and other large mammals preferred at the adult stage [45].

Some studies indicate that forest edges between grass and bush-land is the main risk area for humans to be bitten by a tick. At the landscape level, the connectivity between forest patches has been suggested to play an important role for the movement of larger animals [46].

At the local scale, the two diseases discussed here seem to have slightly different distributions and prevalence: ticks carrying *borreliae* are more widely spread than the TBE virus-infected ones, which are more concentrated in focal areas [47,48,49]. This difference in spatial prevalence pattern at the local scale indicates that the spread and maintenance of the two diseases differ in some aspect related to the ecology of the disease, not the vectoring ticks per se. Factors related to the local habitat structure, species composition, and microclimatic conditions could be decisive. Accessibility (connectivity and fragmentation) and suitability (food resources, shelter, and microclimate) influence the presence of reservoir and competent vertebrate hosts and their dynamics and, thus, the basic resources for the *Ixodes spp.* tick vector and the prevalence of the disease agent [50,51]. For Lyme borreliosis, bank voles and ground-feeding birds (e.g., thrushes), *inter alia*, are competent hosts [43]. Large mammals are very important for tick reproduction, but this does not amplify the disease agents [14,16,22]. For TBE, wood and field mice, bank voles, and shrews are competent hosts whilst large mammals that may also carry the virus are important for reproduction [52,53].

### 2.3. Borreliosis and Tick-Borne Encephalitis as Two CSIs

As an ectothermic arthropod, the *Ixodes spp.* tick’s life cycle and activity depends on the immediate environment’s microclimatic conditions. Ticks depend on their vertebrate hosts for both survival and transport, although a major part of their life cycle is spent away from their vertebrate hosts. Therefore, the local weather conditions and microclimate on the ground influence their activity, development time, survival, and reproductive success [18,41]. Tick vectors are thus likely to be both directly and indirectly affected by the changing climate, and diseases such as tick-borne encephalitis and Lyme borreliosis can be considered climate-sensitive infections (CSIs) [54].

Lyme disease or borreliosis (BOR) is caused by infection by spirochaetal bacteria in the bacterium complex *Borrelia burgedorfi sensu lato*, including several bacterial genospecies that can be transmitted to humans by ticks, mostly from the tick genus *Ixodes*. The different bacterial genospecies have partly overlapping competent vertebrate hosts, such as rodents and birds [55,56] as well as a partly overlapping geographic distribution range of their tick species vectors [57].

Tick-borne encephalitis (TBE) is caused by a flavivirus that is mainly transmitted to humans by ticks infested when taking blood meals from competent vertebrate reservoir hosts, mainly small mammals [48]. There are three subtypes of the TBE-virus: the European subtype, which is mainly transmitted by *Ixodes ricinus*, and the Siberian and Far Eastern subtypes, which are transmitted by the *Ixodes persulcatus* tick [49]. Humans may also become infected by consuming unpasteurised milk products from goats, sheep, and cows, which is a recurrent concern in areas with established tick populations and dairy production from domestic animals considered sentinel hosts for TBE virus (TBEV) [58,59].

## 3. Results

### 3.1. Patterns of Incidence

When studying CSIs, both levels of incidence and trend patterns are relevant as a small increase in a given area may reflect a ramping expansion that may lead to future higher levels of infection in this area. In each district, the trends in BOR and TBE incidence (of the available years over the 1985–2015 period), and the sum of the two incidences (BT) are assigned to one of five groups: well-increasing (WI), increase–plateau (IP), increase–decrease (ID), plateau (P), and decrease–plateau (DP). Figure 1 provides representative examples for BOR, and the spatial distribution of the different groups is shown in Figure 2.

The incidence of BOR is usually higher (overall mean incidence of 7.11 per 100,000, excluding the district of Ahvenanmaa (Finland), which has a mean of 1253 per 100,000) than for TBE (1.7 per 100,000) (Appendix B)). For BOR, 75% of the districts fall into the well-increasing (WI) and increase–plateau (IP) groups, suggesting an overall aggravation of risks of infections over the period 1985–2015 (Table 1).

There is less evidence of an overall increasing trend for TBE, but the proportion of districts assigned to the WI group is about double that for the other groups (with none in the decrease–plateau group). Indeed, the average level of incidence in 2008–2015 (including zero-incidence data) is higher than in 1992–1999 by nearly 196% for BOR and 76% for TBE; these relative ratios increase to 266% and 153%, respectively, when only the WI districts for each disease are considered. For districts not in WI, the ratios are 143% for BOR and 30% for TBE. This confirms a posteriori the WI grouping in an attempt to validate the visual classification (see also the Discussion section). Only 13 WI districts are common to both TBE and BOR, i.e., 57% of TBEs and 37% of BORs (see also Figure 2).

What differentiates these groups and their changes is at the forefront of our understanding of the qualitative and quantitative controls of disease incidence. However, the health data are at the district scale, which means that only the district name was recorded for each cases; consequently, when the incidence is high for a particular district, this does not mean that, everywhere in the district, a large number of new cases were recorded and, the larger the district, the less likely you can pinpoint where cases occurred. For example, Franz Josef Land is part of the Arkhangelsk district (Oblast) with a large continental area at the latitude of Finland, as is the much more northern part of Franz Josef Land and Novaya Zemlya (Figure 2 left panel).

### 3.2. Vegetation Cover Associations

Even though all ecosystems are dynamic over time, increased variation in temperature and precipitation may lead to an elevated risk of extreme weather events, which may increase the risk of forest fires, forest windthrow, flooding, *etc.*. These in turn accelerate changes in vegetation cover [60]. However, on a short time scale, the largest impacts on vegetation come from a combination of changed weather patterns and human activities, including changes in land use [2]. Therefore, we first performed a spatiotemporal analysis of land cover changes over the 1995–2015 period using a multiway principal tensor analysis method [38]. Associations between these changes and disease incidence patterns were then investigated using a multiway correspondence analysis [39].

#### 3.2.1. Vegetation Changes

The multivariate spatiotemporal analysis first differentiates districts only on their level and evolution of vegetation (described by generic Plant Functional Types, PFTs) over the 1995–2015 period without knowledge of their history of disease incidence. The multiway method PTA*k* (Principal Tensor Analysis on *k* modes data) [38] is a generalisation of the Principal Component Analysis (PCA) of a matrix (a tensor of order k=2) to a tensor of order *k*, i.e., a *k*-dimensional data table. For k=3, such a data table corresponds to a collection of matrices and can be visualised as a stack of these matrices along a third dimension, e.g., time. Similar to PCA, PTA*k* decomposes the variation in the data (expressed as the sum of squares of the data values) into a sequence of tensors that sequentially capture the dominant patterns of variation in the data. The land cover dataset used in this study consists of a three-way table (69 districts, 21 years, and 12 PFTs) giving the fractional cover as a function of district, time, and PFT. Applying PTA*k* to this dataset shows that 86.26%, 5.83%, and 4.38% of its variability are captured by the first three best principal tensors, respectively.

In Figure 3 and Figure 4, the weights of each triple of components of the first two principal tensors are displayed as a signed relative contribution (CTR) to the variability, which is defined as the squared component weight divided by the average contribution (equivalent to an uniform equal contribution) and multiplied by the sign of the weight (Appendix C). The larger the relative CTR, the greater the contribution of that variable to the component.

The spatial component of the first principal tensor (Figure 3 left) indicates that some districts (the dark green ones, some in Sweden and some in Finland) make much larger contributions to the overall variation in fractional cover than others. Figure 3 (right) indicates that most of this variation comes from needleleaf evergreen trees (TNE) and natural grass (NG) PFTs, the dominant cover types in the region (Table A2). Most other PFTs contribute very little. The time series component in the first principal tensor (Figure 3 middle) expresses a steady decrease in fractional cover for all PFTs overall but only by a small amount. This small decreasing trend is relative to PFT contributions (Figure 3 right). Since this is a fractional cover data, it has the constraint of adding to 100% (over all the PFTs) for each year. Therefore, this overall small decreasing trend is compensated by other terms in the decomposition.

The second principal tensor (Figure 4) accounts for only 5.83% of the data variability and has the same temporal component as the first principal tensor. The most striking feature of this principal tensor is the importance of managed grassland (MG) in particular districts. Since it has a negative weight, it reduces the values in the first principal tensor when the district weight is positive (e.g., dark blue) and increases them when the district weight is negative (e.g., dark red). Broadleaf deciduous shrubs (SBD) and natural grassland (NG) contribute much less (and other PFTs less still), and since they have positive weights, they have the opposite effect, i.e., they decrease the values in red districts and increase them in dark blue ones. The third-best principal tensor (not shown), representing 4.38% of the overall variation, again has the same temporal component as the first principal tensor. Hence, the small steady decrease in fractional cover for all PFTs from the first three tensors has an aggregated contribution reaching 96.47% of the overall variation. It highlights the combined effect of increasing values of needleleaf trees (TNE) and decreasing values of managed grassland in the dark green districts of Figure 3 and, to a lesser extent, the opposite in the dark blue districts of Figure 4. The subsequent principal tensors each contribute less than 0.03% to the overall variation and are not discussed.

This analysis indicates that there is little vegetation change over the 1995–2015 period, but there are large differences between districts and that the most important PFTs are needleleaf trees (TNE), natural grass (NG), managed grassland (MG) and, to a lesser extent, needleleaf evergreen shrubs (SNE) and broadleaf deciduous shrubs (SBD).

#### 3.2.2. Associations with Incidence Patterns

A multiway correspondence analysis [39] of the average vegetation fraction cover *per* disease, quantiles of disease incidence, and per district incidence temporal pattern is used to highlight multiple associations between the vegetation profiles, levels, and patterns of human disease incidence, where for a given area, e.g., a district, a vegetation profile denotes the distribution of fractional cover across the set of vegetation types. The correspondence analysis compares the four-way table of observed average cover across the four categorical variables (PFT, disease, district temporal pattern, and incidence quantile) to the table that would be expected if these four variables were statistically independent, thus highlighting statistical associations between and within the variables.

For this analysis, the five groups of disease incidence temporal pattern (Figure 1) were tested for association with BOR, TBE, and tularemia (TUL) and for the sum of incidences of BOR and TBE (BT), BOR and TUL (BTU), and BOR, and TBE and TUL (BTT). Including variables with the sum of incidences and using TUL as another tick-borne disease, though with additional means of infection, was performed to add a control on the specificity of any association found. As the focus was on patterns of incidence, to be effective, these controls were chosen as other tick-related diseases (e.g., TUL) or a combination of them as encapsulating non-specific exposures, all for which incidences may have followed similar patterns. For example, in Figure 5c, if BTU had been close to TBE in the left bottom corner, the effect described for TBE (see below) would have been not specific.

Note that, as in logistic regression, the associations resulting from the multiway FCAk method must be interpreted as adjusted with and relative to the other variables involved, here within and across the four dimensions: land cover, disease, quantile, and district pattern groups.

To obtain enough districts per group, the decrease–plateau and plateau groups were amalgamated into a single group denoted Plat, and the increase–plateau and increase–decrease groups were amalgamated into a single group denoted IncP. The well-increasing group was unchanged (labelled Well in the plots).

The four-way table gives the observed vegetation cover (averaged over the 1992–2015 period), $p_{ijkl}$, as a function of vegetation type (*i*), incidence level (*j*), disease (*k*), and district group (*l*). The level of disease *j* is described by its 20% incidence quantiles, each labelled by the upper value of the quantile range, e.g., the label 80% corresponds to the 60–80% set of quantiles (Appendix A, Table A1). The decomposition method, Factorial Correspondence Analysis on *k*-modes (FCA*k*), is analogous to a PTA*k* with k=4 but performed on the data given by $p_{ijkl}/\left(p_{i...}p_{.j..}p_{..k.}p_{...l}\right)$, i.e., the ratios of the observed $p_{ijkl}$ and its fitted value under statistical independence and using weights in the analysis (i.e., the $(p_{i...}$, $p_{.j..}$, $p_{..k.}$, and $p_{...l})$). The weighted sum of squares of these data decomposed by the analysis is equivalent to the chi-squared test of independence between the four variables [39].

In Figure 5, the vegetation associations per disease linked to district groups with increasing or decreasing incidence, or incidence quantiles can be identified with a large CTR magnitude, indicating the variables contributing the most. In reading these associations, multiplication of the signs is required to deduce whether an effect is “positive” or “negative” (Appendix C). For example, Figure 5a shows that a high degree of disease incidence (the 80–100% quantile range) is associated with (1) BOR in WI (well-increasing) districts with substantial managed grassland (all of the weightings are negative so their product is positive); (2) BOR in districts with substantial broadleaf deciduous tree cover, where incidence has plateaued or has decreased followed by a plateau (Plat); and (3) TBE in WI districts with significant broadleaf deciduous tree cover. A mid-range disease incidence (the 40–60% quantile range) is associated with substantial managed grassland for BOR in districts in the Plat group and for TBE in districts in the WI group.

Figure 5b indicates that, irrespective of the disease and district group, higher managed grassland and water fractional cover are associated with the 40–60% and 60–80% quantiles of incidence, while lower managed grassland and water are associated with the 0–20% quantile range. However, this positive “correlation” between managed grassland and disease incidence is weak as the 80–100% quantile range is not involved. The association is stronger for managed grassland than water.

From Figure 5c, irrespective of the level of TBE incidence, the IncP districts, i.e., with an increase–plateau or increase–decrease incidence pattern, are associated with lower broadleaf deciduous tree cover than the other groups. Higher broadleaf deciduous tree cover occurs for the IncP districts for BTU (the sum of BOR and TUL incidence), while districts in the Plat group are associated with lower needleleaf evergreen tree cover for TBE but higher needleleaf evergreen tree cover for BTU.

The fourth, fifth, and sixth FCA*k* principal tensors (not shown) represent 8.24%, 7.53%, and 5.78% of the variability, while each of the others expresses less than 3%. The fourth principal tensor is irrespective of the disease and degree of incidence and associates higher broadleaf deciduous tree and water cover with districts in the IncP group, which, for TBE only, is opposite to the effect shown in Figure 5c. This principal tensor also associates higher managed grass cover with the WI districts, so with increasing incidence. Similar to the first principal tensor, the fifth principal tensor expresses associations between both BOR and TBE, and managed grassland and broadleaf deciduous tree cover but irrespective of district group. For all three diseases, the sixth principal tensor again mainly relates high managed grass cover to the WI group, but only for a high degree of incidence (100% quantile).

Therefore, overall, among the districts with steadily increasing incidence, those with the highest degree of incidence are likely to have higher fractions of managed grassland.

### 3.3. Incidence Levels and Vegetation Changes

The analysis in the previous section linked the average cover for each vegetation type with the levels and patterns of incidence for BOR and TBE. However, the distributions of vegetation cover vary considerably, as seen in the boxplots in Figure 6 and Figure 7. Most of the shrub PFTs show that decreasing cover is linked to a higher incidence of both BOR and TBE (Figure 6). Focusing on the median cover, the assocations can be quite strong, e.g., broadleaf evergreen and needleleaf evergreen shrubs with TBE, broadleaf deciduous shrubs with BOR, but some disease quantile ranges show a large spread of fraction cover.

In Figure 7, we again find a link between managed grassland and high levels of incidence for BOR, but for TBE, the strongest link with the higher managed grass cover is for the 40–60% quantile range of incidence. Natural grass cover, which is a discriminatory variable between districts (Figure 3 and Figure 4), decreases as BOR incidence increases but does not exhibit any simple association with the TBE incidence distribution, except for a U-shape association (i.e., slight U-shape pattern of the medians accentuated by long tails towards higher cover values for the extreme quantile ranges and by a long tail toward low covers for the 40–60% quantile range). Natural grass is nonetheless an important contributing variable in TBE regression models that depict multivariate associations (next section).

Surface interpolations were used to analyse the yearly evolution of the potential associations between managed grass and both diseases (Figure 8). For 35 WI BOR districts, a greater managed grass cover is linked with a high incidence in the later years, but this fades away when all 69 districts are considered. For TBE, a large zone of low incidence (light blue) in 1996–2005 occurs for WI districts but not for all districts, indicating a clear bimodal distribution at 4% and 26% of fraction cover for the 23 WI districts, but this is only weakly present for all 59 districts.

The link between high TBE incidence and high broadleaf deciduous tree cover increases with time (Figure 9), though cover around 8% also shows increasing incidence (a bimodal tendency similar to that for managed grassland). Here, the association is not specific to the WI districts but the bimodality is more prominent for WI. A similar link between higher incidence and increasing broadleaf deciduous tree cover is seen for BOR in WI districts but is not evident for all districts. However, around 2013, there was a high incidence for low broadleaf deciduous tree cover (around 2%). A big difference between WI and all districts is that there are no observations of broadleaf deciduous tree cover higher than 11% in the WI interpolation plot.

### 3.4. Regression Models

Regression models with PFT fractional vegetation cover as explanatory variables were developed only when the yearly incidence was non-zero, once on the whole set of districts and once on the WI districts. The models with zero intercept are blind to the year of observation over the 1992–2015 period. Multi-variate linear regressions with both Gaussian and negative binomial model errors were formed, and a stepwise modelling procedure was used to rank the predictors using the Akaike Information Criterion AIC within each model. The quality and comparison of the fits of the final models can be assessed using the AIC, the squared correlation coefficient ($r^{2}$), and the root mean-squared error between the observed and predicted values (rmse) (Table 2). Only qualitative information on the associations is given in Table 2 as this is the primary aim here. All displayed associations are highly significant (*p*-values <0.001). Using the AIC to compare the quality of the models is possible only for models within either all districts or WI districts.

Overall, Gaussian models perform better than negative binomial (lower AIC and rmse, higher $r^{2}$) for BOR and TBE, and the models for TBE give better fits than for BOR. However, Gaussian distributions are not fully appropriate for the observed incidence distributions as these exhibit long right-hand tails. For each disease, the results of the Gaussian and negative binomial models are more similar for the WI districts than for all districts.

For BOR, the association with broadleaf deciduous trees (+TBD) is the first-ranked predictor except for negative binomial regression for all districts, followed by broadleaf deciduous shrubs (−SBD or −−SBD), whilst for TBE, the best predictors are broadleaf deciduous shrubs (−−−SBD, first predictor in the WI districts) then natural grass +++NG (+NG in all districts) for both regression models. Broadleaf deciduous trees (+TBD) and needleleaf evergreen shrubs (+++SNE) have an increasing effect for BOR in the WI districts, but broadleaf deciduous trees have a decreasing effect for TBE. These differences between BOR and TBE remain when regressing using either the WI districts that are common to both diseases or districts that are in the WI group for either of them. As expected, the regressions on the common WI group have similar signs and amplitudes to those reported on the individual WI groups.

### 3.5. Geographic Forecasts

Projections were obtained for each district by combining land surface modelling (LSM) simulations (Appendix B) with negative binomial regression models using both PFTs and climate variables as predictors. Biome PFTs (Table A3) matching generic PFTs in Table 2 are combined with *temperature* (t°), *soil humidity* (soilhum), *snow depth* (snowdep), and *precipitation* (precip) (Section A.3) to establish the model in Table 3. To easily compare coefficients across Table 3, climate variables were re-scaled between 0 and 1 over the period to match the range of the vegetation type fraction cover. All displayed associations are highly significant (*p*-values <0.001).

Compared with models without climate variables (Table 2), the AIC and rmse errors in Table 3 are lower for both diseases and there are improved $r^{2}$. The vegetation covers are the best predictors for TBE, but for BOR, the climate predictors are the best. Temperature (t°_avSum, t°_avSum_1, t°_avWin_1, t°_maxWin) appears more important for BOR than for TBE (low coefficient and last rank for t°_maxSum_1), whilst humidity variables (soilhum_maxSum and soilhum_avSum) are present in both models, though with a clear increasing effect in the TBE model and potential decreasing effect for BOR.

Boreal broadleaf summergreen trees corresponding to broadleaf deciduous trees (TBD in Table 2) remain significant after including climate variables in both models BOR (+25.62) and TBE (−47.28), as temperate broadleaf summergreen trees are related to broadleaf deciduous shrubs for warm boreal biomes (Table 2). Temperate needleleaf evergreen trees, corresponding to evergreen needleleaf trees, show a similar effect to modelling without climate variables in Table 2 for both models but with a lower intensity for BOR.

C3 grass, in agreement with the results for TBE in Table 2, is now in the BOR model. Needleleaf evergreen shrubs, which in Table 2 had a high positive coefficient, is not significant in Table 3. However, the variable temperate needleleaf evergreen trees with a negative coefficient in Table 3 is in agreement with needleleaf evergreen trees in Table 2. Note, for example, that a 1% increase in C3 grass (natural grass) will multiply the TBE incidence by exp(1.7576)=5.75 or +475% and the BOR incidence by 1.21 or +21%, but this increase will mean a decrease of 1% altogether in the other PFTs, with other increases or decreases depending on the signs of the coefficients. Hence, the uncertainty in the fitted incidences coming from the uncertainty of the predictors can be large and difficult to assess.

Concerning the uncertainty from the model itself, relatively high $r^{2}$ in both models implies meaningful associations. The rmse values are ±9.504 extra cases *per*100,000 and ±1.991 extra cases per 100,000, respectively, for BOR and TBE incidences. These values, close to the 70% quantile for BOR and 68% quantile for TBE (Table A1), represent high values, precluding high confidence in the prediction. Therefore, we can only use these models as indicative of increasing or decreasing trends in incidence.

The standard errors (s.e.) of the coefficients are quite large, for example, for t°_avSum, yielding a 95% confidence interval (CI) between 40% and 246% of the incidence. The CIs for all land cover types are much larger. By averaging the standard error of the fitted log incidence, an overall BOR 95% CI would be 74% to 135% of the fitted incidence, so between 26% less to 35% more of each estimated incidence, and for TBE, the 95% CI would be 72% to 138% of the fitted incidence. The 95% CI for BOR in fact ranges from 53% to 85% and from 117% to 187% for the lower and upper bounds, respectively, while for TBE, the corresponding ranges are 57% to 82% and 120% to 175%, respectively.

Similar CIs can be produced for the projected incidences under the RCP4.5 and RCP8.5 scenarios, but here, the uncertainty in the landscape simulation plays an important part. For the lowest standard error quartile, the 95% CI is already 20% to 500% of the BOR incidence, so between a factor of 5 times less or more than the projected incidences, while for TBE, it is a factor of 3.

Simulated projections for BOR and TBE are presented in Figure 10 and Figure 11 at the district level after averaging projections on the spatial grid used by the LSM. The future trends are consistent with the trend already observed up to 2012 for both BOR and TBE.

## 4. Discussion

### 4.1. Is the Vegetation Effect Due to Specific Types, Their Changes over the Period, or Both?

Some minor changes in land cover were detected in the areas where human incidences of the two tick-borne diseases increased during the study period. The main changes over the 1995–2015 period were slight decreases in cover for needleleaf evergreen trees and natural grass. These are the dominant land cover PFTs (Table A2), but 1% change in fractional cover for this large study area might reflect larger fractions at a finer spatial scale in some areas. Other vegetation types that decreased to a lesser extent were managed grass, needleleaf evergreen shrubs, and broadleaf deciduous shrubs.

Changes in vegetation driven by climate warming may occur at a slower rate than could be detected during the studied period and at this coarse spatial scale. Shrubification, which has mainly been reported from arctic areas and higher altitudes in the north [4], might be slow compared with other vegetation changes, e.g., land use change. Changes in the phenology of plants and other organisms were not considered here. In recent years, Arctic ecological systems have been revealed to be more complex than previously understood [29] and the dynamics of these systems may vary substantially in their response to warming climate. These differential spatiotemporal rates of change can have important consequences in elucidating vegetation associations with CSI incidence. For example, increasing disease incidence over a 20-year period may result from a 40-year trend change in some vegetation type modulated by the variations of intensity of climate change, weather changes over a few successive years, and land use change. Shrubification is likely a consequence of the warming climate, but the rate of vegetation change could also be affected by grazing pressure from large herbivores and changes in land use, especially in northern areas [31]. The warming climate may also cause changes in phenology in northern systems that might be important for local variations in biodiversity and species composition [61] and, hence, the distribution of disease hosts and vectors.

The differentiation of districts according to specific types vegetation profiles from the PTA*k* analysis (Figure 3 and Figure 4) shows some overlap with the differentiation of districts with increasing incidence pattern (WI) for both BOR and TBE (Figure 2). The steady change (Figure 3) was very small at the district scale but could be enough at the ecological level to have an impact on the risk of infection. Indeed, the regression models in Table 2, blind to the years of record, identified significant associations expressing both cover profile differences and changes over the period that may reflect changes in the distribution of hosts and vectors but also change in exposure due to human behaviour.

### 4.2. Are the Differences in Vegetation Association between BOR and TBE Quantitative or Qualitative?

Districts with well-increasing (WI) and stable (plateau) patterns of human incidence for BOR and TBE are associated with high covers of managed grass and broadleaf deciduous trees when these districts also had high incidence levels for BOR and medium levels for TBE. A high incidence of TBE was associated with high cover by managed grass for districts with a plateau pattern of incidence. The WI group for BOR also showed a strong positive association with managed grass, irrespective of incidence level. The incidence group with a stable pattern (plateau group) was less associated with managed grass than the WI group but showed a fairly strong positive association with broadleaf deciduous trees. TBE WI districts with high incidence showed higher association with broadleaf deciduous tree cover than with managed grass. However, districts with the Increase-plateau pattern for TBE were associated with low broadleaf deciduous tree cover. Low cover by needleleaf evergreen trees was associated with districts displaying a plateau pattern for TBE.

To a large extent, these different strengths of vegetation associations for the two diseases arise because their distributions for the WI districts only partially overlap (57% of WI for TBE and 37% of WI for BOR are the same districts) as well as the variations in quantiles of incidence within their WI groups. These multiway associations were confirmed when assessing the cover distributions between quantile levels of incidence (Figure 6 and Figure 7) where the WI BOR districts showed increasing incidences with higher managed grass cover. The WI TBE districts showed a peak of incidence for the mid-range of quantiles (bell shape of the overall association). For broadleaf deciduous tree cover, the WI BOR districts showed no variation in incidence over time but no cover higher than 11% (Figure 9) while the WI TBE districts seem to fluctuate a little with an increasing trend over time for fraction cover around 8%.

For natural grassland, the BOR WI districts show a negative trend (both in reduced variation and medians) with higher quantiles while WI TBE incidence medians do not change over quantiles but show a small increase from mid-range quantile to the maximum levels within a U-shaped association (Figure 7). Shrub cover mostly shows a slight decrease with higher quantiles except for the WI TBE group for which the association seems to increase with higher quantiles (Figure 6).

The results from the regression models quantify these semi-qualitative findings about associations between disease incidence and vegetation cover, most of the time confirming them. Therefore, when clear effects were highlighted qualitatively, they were confirmed quantitatively, and some other quantitative findings were also partially supported qualitatively.

Why are these different vegetation profiles related to differences in the diseases? One reason could be that the hosts maintaining the viral disease, TBE, and the bacterial disease, BOR, differ in distribution, local abundance, and competence to harbour and maintain the different disease agents. Two closely related tick species dominate in the east and west, but their distribution ranges overlap in the zone in between [49,62]. Both species are known to be competent vectors of TBE and BOR, but slight differences in life history traits can influence to what degree they become infected [62] and hence transmit the disease to humans.

However, the prevalence of BOR in all districts indicates that one or both tick species are present in some part of all reported districts. TBE incidences are reported for 59 out of the 69 districts with BOR incidence. Ten districts in the west (nine in Norway and one in Finland) have no reports of TBE. The 13 WI districts common to both BOR and TBE are found across Russian districts (2), Finland (7), and some south Sweden areas (4) (Figure 2), without specific east–west delineation. Note the relative absence of WI districts for TBE in Norway (only one, south of Oslo), which could be due to the missing TBE data for nine districts.

### 4.3. Are the Temporal Patterns of Associations Different for BOR and TBE?

Surface interpolations of the WI district incidences for TBE and BOR over time with managed grass cover (Figure 8) show different patterns for the two diseases when compared with the interpolations with all districts. The change in association over the years is therefore due to the specific cover change of the districts with increasing patterns of incidences.

For BOR WI districts, this is a marked increase in incidence from 2007 to 2015 with moderate and large amounts of managed grass, which disappears when all districts are considered. For TBE, a large area of low incidence is linked to a wide range of fractional cover between 1996 and 2005 (Figure 8), but between 2005–2010, a high incidence is linked to a bimodal spread with high and low managed grass cover, around 26% and 3%, respectively. For TBE, high managed grass cover is associated with a high incidence from 2005, with persistent hotspots between WI and all districts. However, a fractional cover for managed grass reaching 26% is relatively high as the average cover is 9%.

For broadleaf deciduous trees (Figure 9), a hotspot of high incidence of TBE for WI can be seen around 7% cover from 2003 to 2012, but this vanishes when all districts are considered. Figure 9, for all districts, shows also a hotspot at much higher cover (24%), which is rare given that the average cover is 5% (Table A2)). Surface interpolations with the other PTFs could not be as clearly interpreted as with managed grass and broadleaf deciduous trees, which illustrates the complexity in the temporal pattern of association.

### 4.4. Is Regression for WI Districts a Way to Obtain Better Fit or More Meaningful Associations?

Whether we consider all districts or only WI districts, Gaussian regression models for both diseases gave a better fit (Table 2). However, the WI district models had higher $r^{2}$ irrespective of whether a Gaussian or negative binomial model was used. Additionally, the difference in $r^{2}$ between Gaussian and negative binomial regression was smaller for the WI samples. The associations for Gaussian and negative binomial regressions were also more consistent for the WI samples. This shows that the relation between relative vegetation cover and disease incidence is stronger when considering districts with an increasing incidence over the period (WI districts). As incidences are better linked to vegetation variables for the regression with the WI districts and as high incidence values are more likely to occur towards the end of the studied period by construction of the WI, a temporal effect underlies these associations. Hence, some increase or decrease in vegetation may be linked to the period of observations and be responsible for the differences between the regressions for all districts and WI districts.

Therefore, a mixture of vegetation profile and small vegetation changes within the WIs are likely to be driving the increases in incidence. Figure 6 suggests a negative correlation between broadleaf deciduous shrubs and BOR incidence but a positive correlation with TBE. The regressions (Table 2) confirm this for BOR but indicate a very significant opposite effect for TBE. The boxplot for TBE in Figure 6 had nonetheless a large spread of fraction cover for the lowest disease quantile, with higher values than for the highest disease quantile, which must have contributed to a negative effect in the regression. Natural grass and TBE links in Table 2 and Figure 7 are consistent.

For the WI districts, BOR and TBE models do not share the same important vegetation variables and sometimes indicate opposite effects. Broadleaf deciduous shrubs, broadleaf evergreen shrubs, needleleaf evergreen trees, mosses and lichens, and needleleaf deciduous trees had similar effects for BOR and TBE, while broadleaf deciduous trees, sparse vegetation, and managed grass had opposite effects (Table 2). This could partly be due to the different sets of WI districts for the two diseases (Table 1). The regressions on the sample of WI districts common to both BOR and TBE showed similar effects to those found on their respective WI districts (though with different ranking and strengths).

When both biotic and abiotic factors are considered (Table 3), the main difference between the incidences of BOR and TBE is that the first three explanatory variables for BOR are *t° summer average* and *t° last year summer average*, with an increasing effect, and *soil humidity* (negative effect), while the TBE model has five vegetation variables, of which four have a negative coefficient (all except natural grass). The gains relative to models with only vegetation are substantial (BOR: −645 in AIC and +5% in $r^{2}$, TBE: −411 in AIC and +8% in $r^{2}$). In addition, models with climate variables alone (not shown) give a better model fit than for vegetation alone (BOR: −553 in AIC, TBE: −310 in AIC) and lower least squares values (BOR: −3% in $r^{2}$ and TBE: −7% in $r^{2}$).

### 4.5. Do the Projections Reflect Real Trends or Increasing Uncertainty?

The models including both vegetation and climate variables were used with climate-driven land surface simulations for projections up to 2070. However, with fitted values explaining only 56% and 66% of the variability for BOR and TBE, respectively, and the estimated average standard error having a confidence around ±30% of the estimated incidences, any forecasts have to be treated with caution. The accuracy in forecast also depends on the uncertainty attached to the landscape forecast and the climate forcing uncertainty, including potential changes in their covariance. The impact of this uncertainty was seen in the forecasted CI from the model to levels at least five times less or more for BOR and a factor 3 for TBE. However, the predicted trends are qualitatively consistent with the trends already observed for BOR and TBE up to 2012.

For BOR, an eastern increase (southwest of Finland and northwestern Russian districts) intensifies up to 2050, reaching the middle of Sweden and southern Russian districts. A hotspot around Oslo occurs up to 2040. Between 2050 and 2070, a decrease over this zone still leaves high incidences in the middle of Sweden, west of Helsinki and the Russian district of Vologda. For TBE, the increasing trends in Sweden and western Russia also intensified up to 2060 but with more heterogeneity. The Norway projections are different from the observed period, as Norway had districts with missing data. The intensification in the southern districts of Norway is nonetheless consistent with the observations.

The negative binomial distribution model was found more appropriate than a Gaussian model due to the long right-hand tails for BOR and TBE incidences; hence, any uncertainty in predictor values has more impact on prediction accuracy due to the exponential function.

### 4.6. Limitations of This Study

The classification of the districts based on the raw time series of incidence and their smoothed versions is subject to uncertainty. The districts were visually assigned to one of the five classes of simple patterns: well-increasing (WI), increase–plateau (IP), increase–decrease (ID), plateau (P), and decrease–plateau (DP). The simple patterns are obviously crude approximations of the historical incidences, and some districts may be more difficult to assign than others. A visual assignment has not only some benefits from taking into account a complex perception process but also, as a consequence, limitations concerning validation (e.g., influence of the assessor and full comprehension of the subjective criteria used). Attempting to quantitatively validate the grouping is becoming challenging. The time series has 21 time points and is very irregular with the presence of small outbreaks, precluding acceptable statistical inference with confidence intervals around the smoothed line. Nevertheless, as an a posteriori validation, the patterns correspond at the group level for both BOR and TBE, at least for the WI group, as described by the average relative increases from 1992–1999 to 2008–2015 (Section 3.1 after Table 1).

It is important to be cautious when interpreting the results of a study at this coarse scale and with uncertainties in the response and explanatory variables used. As the health data were at the district scale, the approach taken here was to upscale the potential predictors from abiotic and vegetation data at the district level. A downscaling approach of the health data to a finer spatial scale than districts could provide further understanding of these results and the quality of the projections of the future development of these diseases. However, downscaling, for example according to population density, could also generate uncertainties as human mobility within a district can be linked to contracting the diseases. Focusing on areas where field studies of tick distributions, their vertebrate hosts, and tick-borne diseases have been performed could significantly enhance our understanding of the role of vegetation changes. For example, investigating the role of fragmentation and mixture of habitats at this finer scale could yield important potential predictors related to the hosts.

Several recent studies stress that the understanding of ticks, their basic ecology, and the epidemiological dynamics of tick-borne diseases need to be improved. Our findings highlight this incomplete view and that field studies at different spatial scales in different ecological environments combined with cutting edge techniques in molecular biology and microbiology as well as laboratory studies under controlled conditions should be encouraged [63].

Differences and similarities in vegetation associations between Lyme borreliosis and tick-borne encephalitis for the reporting districts with patterns of increasing incidences may also indicate areas where humans work or spend their leisure time, and then are bitten by ticks. These areas, with a mixture of deciduous woodland and managed grass, are suitable for the tick hosts. The differences in vegetation associations for Lyme borreliosis and tick-borne encephalitis indicated here may depend on differences in the ecology and epidemiology of the disease agents per se and vertebrates are the main means of tick dispersal and maintain the disease agents.

## 5. Conclusions

This paper investigates whether coarse-scale composition and potential changes in vegetation cover could be among the driving factors of the increasing human incidence of two tick-borne diseases observed during recent decades. Only small changes in vegetation cover were detected at the scale of the entire study area, but the large variation in district size and changes in vegetation cover at the district level might dilute the substantial changes in some districts.

The incidence rates for Lyme borreliosis and tick-borne encephalitis, for the districts with increasing patterns over the 1985–2015 period, were associated with different vegetation types in addition to the known effects of abiotic factors [15,18]. The effects of different vegetation types, land use, and landscape factors on tick distribution and tick-borne diseases have been shown in studies [16,51] performed at a much finer spatial scale than the current study.

Separate regression models for human incidence against climate and vegetation cover variables indicate different weights for the abiotic and biotic explanatory variables for the two diseases. In districts with a pattern of increasing Lyme borreliosis, three abiotic variables—*t° summer average* and *t° last year summer average*, with increasing effects, and *summer average soil humidity*, with a reducing effect—were the most important in explaining the variation in incidence, followed by the *deciduous forest* and *natural grass* vegetation variables. Tick-borne encephalitis incidence in districts with an increasing pattern of incidence was primarily explained by five vegetation variables: *natural grass* (with a larger coefficient than for Lyme borreliosis), *deciduous forest* (reducing effect), and *conifer forest* (reducing effect) being the first three variables and then *boreal deciduous forest* and *managed grass* with reducing effects. The next most important explanatory variables were abiotic: *maximum summer soil humidity* and *summer average soil humidity* (both with increasing effects). For both diseases, *broadleaf shrubs* were associated with reduced incidence but these relations vanished when taking into account abiotic factors. Using multiway descriptive methods, vegetation associations were found with other incidence patterns, such as *broadleaf deciduous forest* with districts with plateau patterns of tick-borne encephalitis and *conifer forest* with districts with increasing plateau patterns of tick-borne encephalitis. For both diseases and all district patterns of change of incidence, *managed grass* was associated with higher levels of incidence.

The differences between the two diseases cannot be explained by the fact that two different tick species dominate in the eastern and western parts of the study area for two main reasons: (1) both species are known to transmit the diseases, and human incidence of Lyme borreliosis is recorded in all 69 districts, so it is probable that at least one of the tick species is present in some part of each district; (2) the districts with a pattern of increase are distributed over the entire study area.

Therefore, the observed differences in the importance of abiotic and biotic explanatory variables in areas of increased disease incidence could be attributed to the ecology of the diseases, such as local interaction between ticks and their vertebrate hosts [64], and where and when humans are exposed to ticks.

## Figures and Tables

**Figure 1 ijerph-18-10963-f001:**
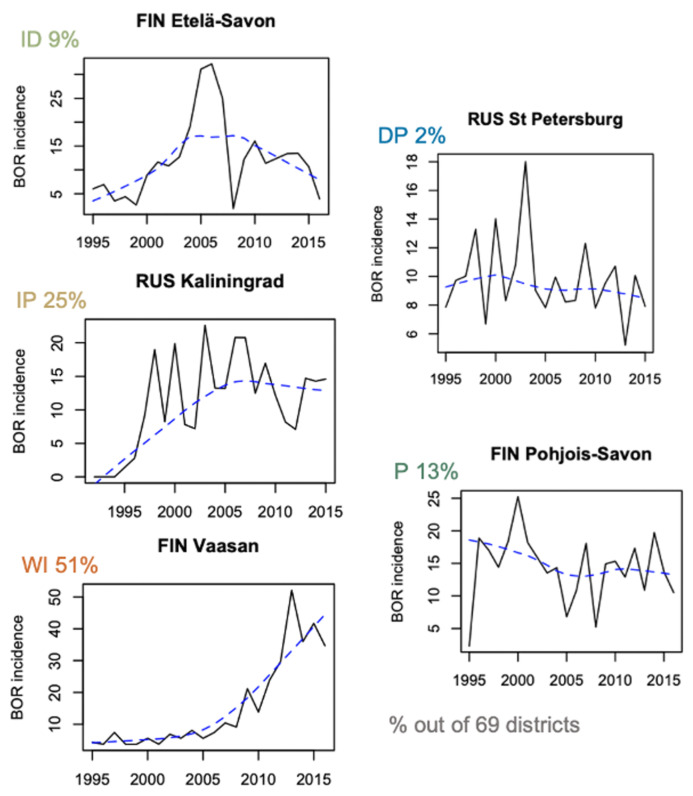
Representative examples of the time series of Lyme borreliosis incidence in districts assigned to the well-increasing (WI), increase–plateau (IP), increase–decrease (ID), plateau (P), and decrease–plateau (DP) groups; the percentages of districts assigned to each group are also indicated; the black line joins the yearly data, and the dotted blue line is the non-parametric local regression smoothed line (loess).

**Figure 2 ijerph-18-10963-f002:**
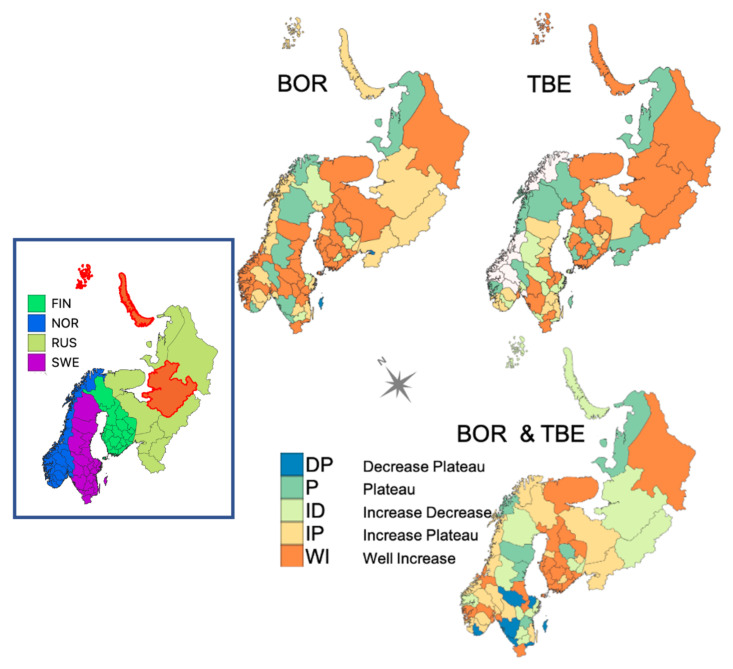
Districts coloured according to the five groups of incidence trend for Lyme borreliosis (BOR), tick-borne encephalitis (TBE), and their combined incidence (BT) over the 1985–2015 period (**right panel**); districts coloured according to their countries with a highlight in red for the Russian district of Arkhangelsk in three parts (**left panel**).

**Figure 3 ijerph-18-10963-f003:**
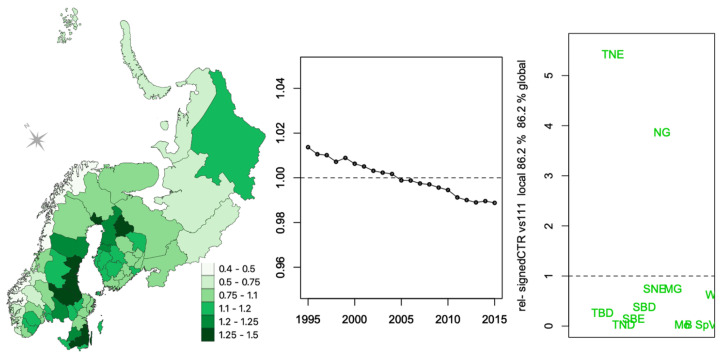
Best principal tensor representing 86.26% of the variability in vegetation fractional cover per 69 districts × 21 years × 12 PFTs. The plots show the spatial (**left**), temporal (**middle**), and vegetation PFT (**right**) component weights of the relative signed CTR (a score of ± 1, the uniform equal contribution, is indicated by the horizontal dashed line on the two figures on the right-hand side). The horizontal spread in the one-dimensional PFT plot is used simply to clarify the display; see Appendix A for the PFT acronyms (Table A2).

**Figure 4 ijerph-18-10963-f004:**
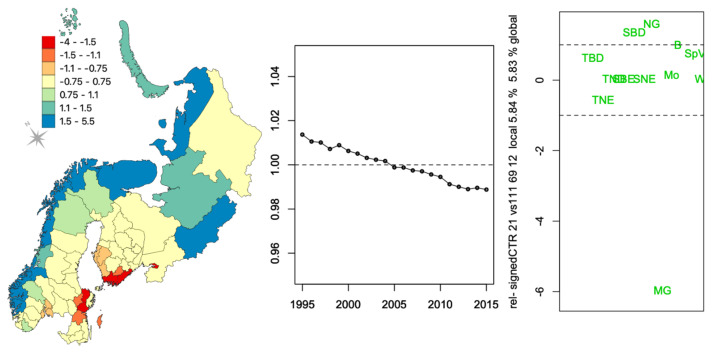
Second best principal tensor representing 5.83% of the variability in vegetation fractional cover per 69 districts × 21 years × 12 PFTs. The plots show the spatial, temporal, and vegetation (PFTs) component weights of the relative signed CTR (a score of ± 1, the uniform equal contribution, is indicated by the horizontal dashed line on the two figures on the right-hand side). The horizontal spread in the one-dimensional PFT plot is used simply to clarify the display; see Appendix A for the PFT acronyms (Table A2).

**Figure 5 ijerph-18-10963-f005:**
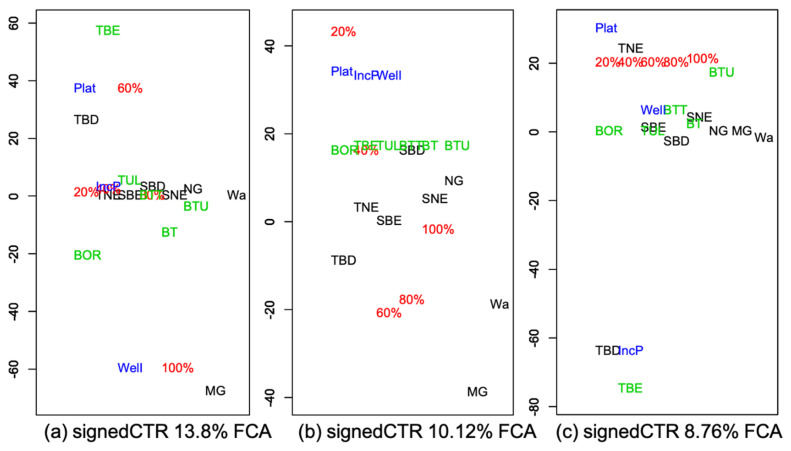
The three best FCA*k* principal tensors representing (**a**) 13.8%, (**b**) 10.12%, and (**c**) 8.76% of departure from independence. The four components of each principal tensor are all shown on the same one-dimensional plot and indicated by different colours (black = *vegetation type*, red = *incidence quantile*, green = *disease*, and blue = *district group*), with the signed-CTR as vertical coordinate. The horizontal spread in each plot is used simply to clarify the display.

**Figure 6 ijerph-18-10963-f006:**
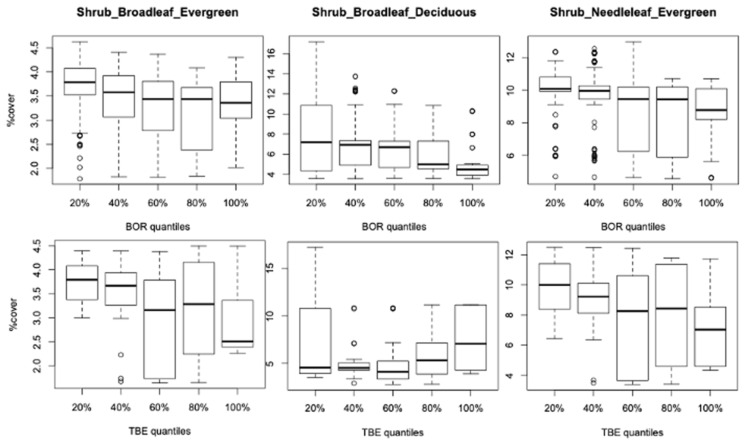
Boxplots for the shrub cover fractions for the WI districts per quantile range of disease incidence.

**Figure 7 ijerph-18-10963-f007:**
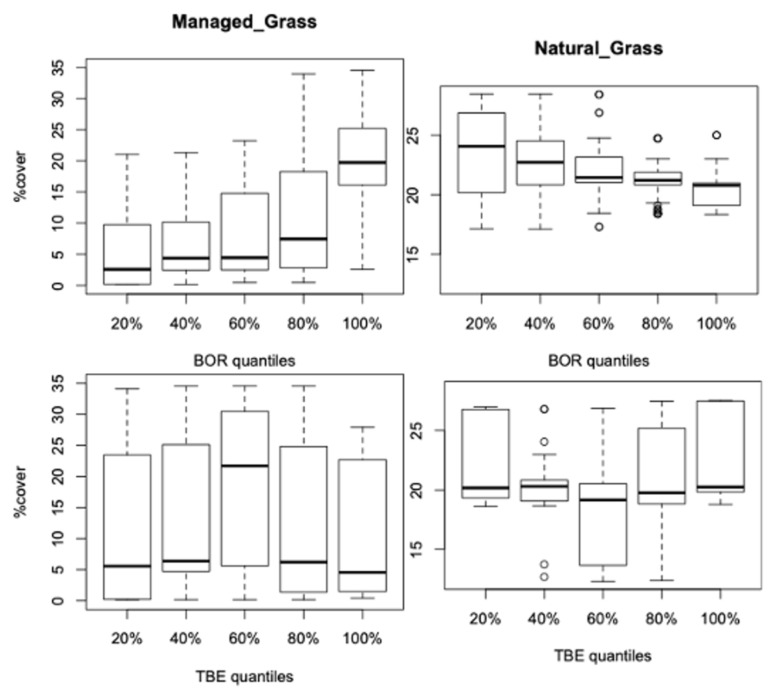
Boxplots for the managed grass and natural grass cover fractions for the WI districts per quantile ranges of disease incidence.

**Figure 8 ijerph-18-10963-f008:**
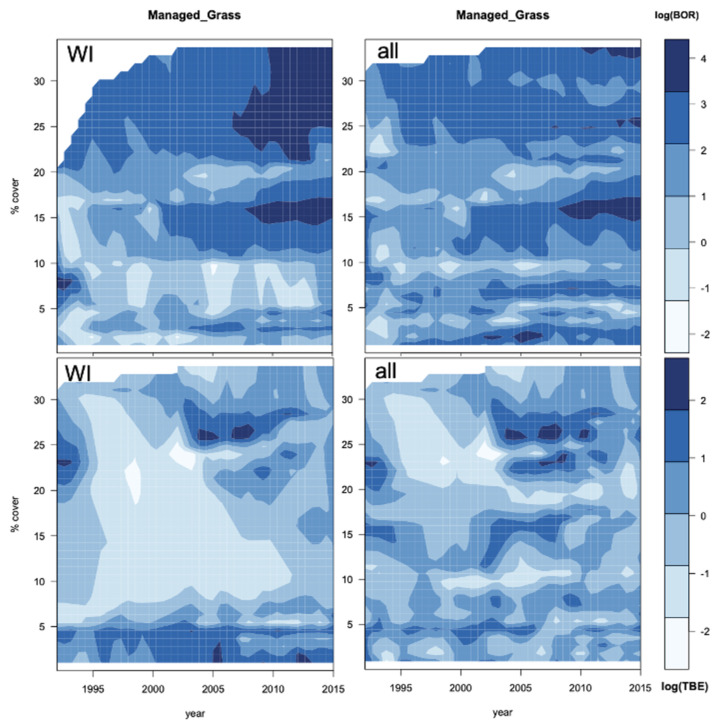
Interpolations for the log of incidence of BOR (top) and TBE (bottom) with respect to the percentage cover of yearly evolution of managed grass for (**left**) the WI districts and (**right**) all districts.

**Figure 9 ijerph-18-10963-f009:**
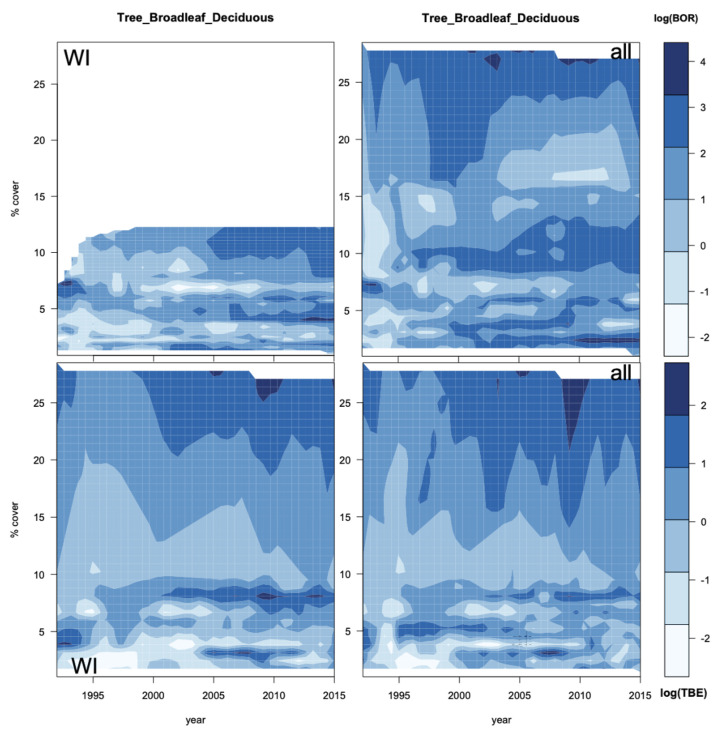
Interpolations for the log of incidence of BOR (**top**) and TBE (**bottom**) with respect to the percentage cover of yearly evolution of broadleaf deciduous trees for (**left**) the WI districts and (**right**) all districts.

**Figure 10 ijerph-18-10963-f010:**
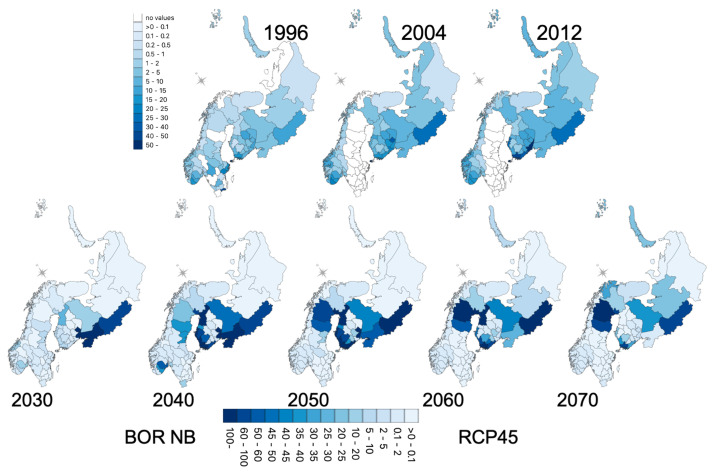
Lyme Borreliosis (BOR) incidence: (**top**) historical period, seven-year averages around each reported year; (**bottom**) projected incidences (seven-year averages) after negative binomial modelling using vegetation types and climate variables, and RCP4.5 land surface model simulations.

**Figure 11 ijerph-18-10963-f011:**
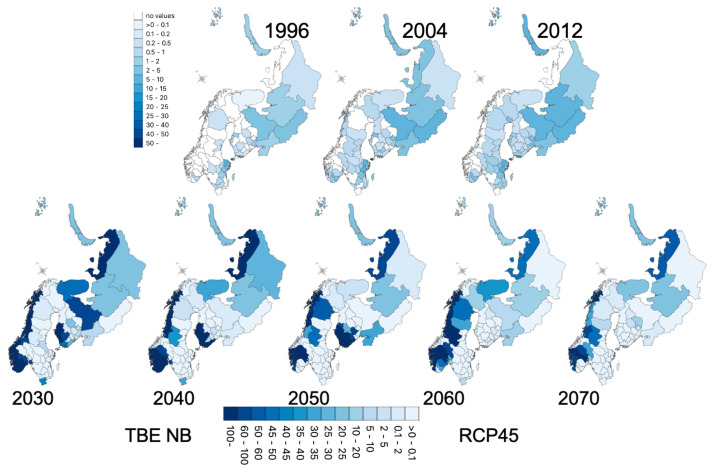
Tick-borne encephalitis (TBE): (**top**) historical period, seven-year averages around each reported year; (**bottom**) projected incidences (seven-year averages) after negative binomial modelling using vegetation types and climate variables, and RCP4.5 land surface model simulations.

**Table 1 ijerph-18-10963-t001:** Distribution of incidence pattern groups for BOR (rows), TBE (columns), and total incidence (BT) (10 districts in Norway had no TBE reports).

		TBE						
		1	2	3	4	5	na	BOR (N = 69)
**BOR**	**Decrease–Plateau 1**	0	1	0	0	1	0	**2**
	**Plateau 2**	0	3	2	1	2	1	**9**
	**Increase–Decrease 3**	0	2	0	2	2	0	**6**
	**Increase–Plateau 4**	0	2	3	3	5	4	**17**
	**Well-Increasing 5**	0	4	7	6	13	5	**35**
	**TBE (N = 59)**	**0**	**12**	**12**	**12**	**23**	10	
	**BT (N = 69)**	**7**	**6**	**12**	**19**	**25**		

**Table 2 ijerph-18-10963-t002:** Qualitative levels and signs of the coefficients for generalised linear regression models for BOR and TBE incidences with PFT fractional cover as explanatory variables; +++ indicates a high and positive coefficient, ++ indicates a less high coefficient, −−− indicates a high negative coefficient, etc. The acronyms for the PFTs are explained in Table A2.

	All Districts				WI Districts			
	BOR		TBE		BOR		TBE	
Rank	Gaussian	Neg Bin	Gaussian	Neg Bin	Gaussian	Neg Bin	Gaussian	Neg Bin
1	+TBD	+MG	+NG	+MG	+TBD	+TBD	−−−SBD	−−−SBD
2	−−TND	+NG	−−SBE	+++TND	−−SBD	−SBD	−−TNE	++Spv
3	+++SBE	−−SBD	+Mo	++SNE	−−Spv	+MG	+++NG	+++NG
4	−−SNE	+TND	−Spv	+NG	−−−TNE	−−−TNE	−MG	−TNE
5	−SBD	+++SBE	−SBD	+Mo	++++SNE	+++SNE	−−TBD	−MG
6	+MG	++Spv	+MG	−−SBE	−−−Mo	−−Mo	−−Mo	−TBD
7	−Spv	−−SNE			−−−SBE	−Spv	−−−SBE	−Mo
8	++Mo	+Mo			+++TND	−−−SBE	++TND	
AIC	6909	18909	2326	8895	3524	9405	1066	4327
$r^{2}$	49%	21%	50%	26%	58%	51%	68%	58%
rmse	10.28	28332	2.35	3664	8.85	9953	1.9	2215

**Table 3 ijerph-18-10963-t003:** Negative binomial regression models for BOR and TBE incidences with biome PFTs and climate as explanatory variables over the whole 1992–2015 period. (Biome PFTs have been partially abbreviated: temp is for temperate, broad is for broadleaf, and needle is for needleleaf).

	Neg Bin Regressions on WI Districts $\log\left(y\right)=\sum_{i}\beta_{i}x_{i}$ and y∼NB()			
y	1000 × BOR		1000 × TBE	
Rank	$x_{i}$ Coefficients $\beta_{i}$ (s.e.)		$x_{i}$ Coefficients $\beta_{i}$(s.e.)	
1	t°_avSum	4.89 (0.45)	C3_grass	175.7 (18.7)
2	soilhum_avSum	−3.78 (0.38)	temp_broad_summergreen	−313.1 (30.6)
3	t°_avSum_1	3.89 (0.42)	temp_needle_evergreen	−115.2 (13.9)
4	boreal_broad_summergreen	25.6 (3.58)	boreal_broad_summergreen	−47.3 (7.65)
5	C3_grass	19.6 (2.82)	C3_agriculture	−17.1 (2.72)
6	t°_avWin_1	2.05 (0.34)	soilhum_maxSum	7.13 (1.44)
7	temp_needle_evergreen	−16.7 (2.93)	soilhum_avSum	2.34 (0.45)
8	moss_lichen	−19.5 (3.99)	precip_max	−1.83 (0.43)
9	precip_avSum	2.54 (0.47)	moss_lichen	−37.0 (10.8)
10	soilhum_maxSum_1	−1.79 (0.40)	C3_grass_arctic	−56.0 (18.4)
11	t°_maxWin	−1.13 (0.31)	t°_maxSum_1	1.02 (0.51)
12	boreal_needle_deciduous	−59.2 (17.3)		
13	precip_av_1	1.77 (0.62)		
14	soilhum_maxSum	2.25 (0.98)		
15	boreal_needle_evergreen	−2.58 (1.25)		
AIC	8760.6		3916.2	
$r^{2}$	56%		66%	
rmse	9504.4		1991.8	

## Data Availability

The data presented in this study are openly available in the Zenodo repository at doi:10.5281/zenodo.5572309, reference number 5572309.

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
