# Peer review of "Associating Land Cover Changes with Patterns of Incidences of Climate-Sensitive Infections: An Example on Tick-Borne Diseases in the Nordic Area"

_ijerph, 2021, doi:10.3390/ijerph182010963_

Round 1

Reviewer 1 Report

Leibovici et al.:  Associating land cover changes with patterns of incidences of climate sensitive infections: an example on tick-borne diseases in the Nordic area.

The authors attempt to identify connections between recent environmental changes in the boreal and arctic Europe and concurrent changes in the incidence of tick-borne diseases. In spite of an appreciable analytical effort, the results are inconclusive (e.g. natural/managed grass - identified to be associated with the diseases - is a biotope largely avoided by the ticks in question). Apart from exceptionally convoluted connections, shortcomings in employing the epidemiological data may be responsible:

1/ The data should have undergone a critical revision prior to the analysis in order to eliminate potential artefacts. In particular, the areas where acquisition of tick-borne diseases is quite unlikely should have been excluded. For example, there could hardly be recorded any autochthonous disease cases in Novaya Zemlya and (unpopulated) Franz Josef Land – still, these areas are included in the analysis which looks weird (p.6, Fig.2).

2/ Each disease represents a peculiar entity as to its biology, epidemiology, and reporting, and their distribution data cannot be soundly mixed with one another (p.21, l.626-23). What extra information emerges if the data are mixed together compared to if they are considered in isolation?

3/ I basically like the authors’ idea to classify the geographical areas according to a historical trend in the disease incidence (overall increase, decrease…), and to simplify thereby the analysis. What I cannot understand it is that the authors made it only by eye (p.3, l.97-100) when it determines the analysis precision. To check reliability of the authors’ visual approach in some objective way, I used the time series shown in Fig,1 (p.5) and tentatively estimated confidence limits of ‘white noise’ (via bootstrapping, using the procedure ‘tsboot’ of R, and ‘loess’ adjusted to come as close as possible to the trends shown in Fig.1). It appears that only two of the five classifications exemplified in Fig.1 are formally valid (namely ‘IP’ and ‘WI’). The other supposed trends either run within 95% confidence limits of ‘white noise’ and can be attributed to chance only, or go beyond the limits while the opposite is expected (‘P’). Based on this example, I have strong reservation over adequacy of the authors’ approach and question validity of the conclusions drawn. Each trend should have been tested against a random model, not assigned subjectively!

The manuscript is wordy and poorly organized. The authors jump between methods, discussion, and other matter. For example, the matter in the paragraph 2.1, inclusive of the remark text box on p.4, belongs to Introduction or Discussion, not to M&M. On the other hand, methodological matter is impractically dislocated in a couple of appendices at the end of the article - the reader needs constantly to leaf through and fro… The Conclusions section, that - according to Instructions for authors – “provides readers with a BRIEF summary of the main conclusions”, is over a full page long and resembles rather another paragraph of Discussion including citations. Etc., etc…

Altogether: I regret I cannot recommend this article for publication in its present form. My suggestion is rejection and re-submission after a thorough revision and abridgement. 

Reviewer 2 Report

The manuscript describes a phenomenon of increase in the incidences of Lyme disease and Tick-Borne-encephalitis (TBE) in northern Europe and western Russia with change in vegetation pattern. The manuscript is lengthy as well as there are few flaws which are listed below. 1. English: there are several mistakes in English, which need to be corrected. 2. In figure legend, explain the line. In figure 1 there are two lines, black and blue. Black looks like the values, and blue appears to be the trend line. Would you please explain them in the legend? 3. Would you please describe the numbers written on the top of each figure? For instance, what ID 9% or IP 25% means in the figure legend. 4. Figure legends should be detailed, so a person should not read the entire paper to understand them. 5. In Line 171: “overall mean incidence of 7.11/100 000” means. 7.11 cases per 100 000 people, please describe them. 6. Figure legend for Figure 3 should be broken into three sections to describe them better. 7. In figure 6, since your x and y axes are common for all of the figures, please use common axis labels. 8. Figure 7: Please change the axis as described in Figure 6 9. Figure 8 shows an increase in BOR between 2010 and 2015, despite any changes in cover. The same holds for TBE. 10. In figure 10, the top of figures (1996-2012) and bottom figures (2030-2070) use two different color legends. Please separate them as (a) and (b) for easy interpretation. 11. In figure 11, the top of figures (1996-2012) and bottom figures (2030-2070) use two different color legends. Please separate them as (a) and (b) for easy interpretation.

Reviewer 3 Report

The submitted manuscript presents the results of analyses performed to study  spatiotemporal changes in human incidence of two tick-borne diseases caused by Borrelia burgdorferi sl and Tick-Borne Encephalitis virus. By combining with climatic factors, the study investigates whether changes in land cover and vegetation type in Fennoscandia and Western Eurasia, could explain the distributions of the disease prevalence by linking those changes to the extended distribution range of ticks and their hosts.

Overall, the manuscript is well written, the analyses are adequately designed and performed, and the results obtained are statistically well substantiated and support the hypothesis that land conditions, including vegetation types and cover, within recent and foreseen climatic changes constitute a set of potential drivers of changes in disease prevalence. Thus, changes in vegetation type in Fennoscandia and western Eurasia, act as potential predictors linked to habitat suitability for ticks and their hosts, which could explain the distributions and incidences of borreliosis and tick-borne encephalitis, that have steadily increased over the 1995-2015 period. It is shown in the study that by combining vegetation cover and climate variables in regression models, the interplay of biotic and abiotic factors is linked to Borreliosis and tick-borne encephalitis virus incidences, and identifies some differences between both diseases. Regression model projections up to year 2070 under different climate scenarios depict possible disease progressions within the studied area and are consistent with changes observed over the past 20 years.

A few minor items worth of revision were found by the reviewer and authors are invited to correct them in order to improve the quality of the submitted manuscript:

Line 43. Change “lato sensu” to “sensu latu”; and “virus disease” to “viral disease”

Line 46. Correct “vector other zoonotic” to vector of other zoonotic”

Line 49. Correct “Ixodes ricinis” to “Ixodes ricinus

Line 55. Change “virus disease” to “viral disease”

Line 56-60. Please check if references 23 and 24 cited in the text are not the same, as reference 23 should be correctly typed in the reference section as follows:

  1. Jia, G.; Shevliakova, E.; Artaxo, P.; De Noblet-Ducoudré, N.; Houghton, R.; House, J.; Kitajima, K.; Lennard, C.; Popp, A.; Sirin, A.; Sukumar, R.; Verchot, L. Land–Climate Interactions. In: Climate Change and Land: an IPCC special report on climate change, desertification, land degradation, sustainable land management, food security, and greenhouse gas fluxes in terrestrial ecosystems [Shukla, P.R., Skea, J.; Calvo Buendia, E., Masson-Delmotte, V., Pörtner, H.-O., Roberts, D.C., Zhai, P., Slade, R., Connors, S., van Diemen, R., Ferrat, M., Haughey, E., Luz, S., Neogi, S., Pathak, M., Petzold, J., Portugal Pereira, J., Vyas, P., Huntley, E., Kissick, K., Belkacemi, M., Malley, J. (Eds.) ]. In press. 2019

As reference 23 appears to be exactly the same as reference 24, the latter should be eliminated. This, however, would change the citation numbering in the text and consequently, all subsequent references in the references section.

Lines 130-157. It is not clear for the reviewer why a text box containing information on the two diseases subject of the study (Borreliosis and tick-borne encephalitis, two CSIs) is included in the materials and methods section. Shouldn’t this information be part of the introduction section?

All scientific names should be typed in italics. Please check throughout the manuscript for all scientific names that need to be typed in italics. For example: Borrelia in lines 138, 777, 851, 865, 870 and 889.

Line 704. Reference 1. This chapter should be correctly cited as follows:

Meredith, M.; Sommerkorn, M.; Cassotta, S.; Derksen, C.;  Ekaykin, A.; Hollowed, A.; Kofinas, G.; Mackintosh, A.; Melbourne-Thomas, J.; Muelbert, M.M.C.; Ottersen, G.; Pritchard, H.; Schuur, E.A.G. Polar Regions. In IPCC Special Report on the Ocean and Cryosphere in a Changing Climate [Pörtner, H.-O., Roberts, D.C., Masson-Delmotte, V., Zhai, P., Tignor, M.,  Poloczanska, E., Mintenbeck, K., Alegría, A., Nicolai, M., Okem, A., Petzold, J., Rama, B., Weyer, N.M. (Eds.)]. In press. 2019

Line 708, reference 2. Delete: “Number: 7720 Publisher: Nature Publishing Group”

Please check and delete similar descriptors in all article references that contain a Publisher name and type of article, for example: Reference 3 “Publisher: American Association for the Advancement of Science Section: Review,”

Line 719-720, reference 3. Use Abbreviated Journal Name (Glob Change Biol). Delete:  _eprint: https://onlinelibrary.wiley.com/doi/pdf/10.1111/j.1365-2486.2009.01935.x,

Check all article references and make the following changes:

Some article titles contain all nouns typed with capital letters (see, for example, reference 17). Except for the first one, they all should be in small caps. Check and correct all references presenting with the same situation. Use abreviatted Journal names, delete Publisher`s names.

Check for scientific names in titles and use italics (see for example Ixodes ricinus in reference 13, and Ixodes persulcatus in reference 14).  

Line 954, reference 68. Add names of missing authors (others)

Round 2

Reviewer 1 Report

Leibovici et al.:  Associating land cover changes with patterns of incidences of climate sensitive infections: an example on tick-borne diseases in the Nordic area. / 1st revision

I acknowledge partial improvements mainly in the Discussion section. I’m still not convinced that the author’s approach is optimum, but understand the difficulties with an intervention into the methodology at this stage of the publication process. Concerning my previous comments and authors’ replies:

1/ Well, the authors are limited by the data-set at their disposal (even though to mask out an unwanted area from the source land-cover must be a matter of one or two GIS operations at the most…), why then the artefacts are not, at least, erased from the final maps (particularly if the author’s attitude is “little areas have little influence…”)?  A perceptive reader will easily notice e.g. the absurdity of TBE in Franz Josef Land and Novaya Zemlya (Fig 2.), and quickly conclude that something is wrong with the data.  Is it so difficult to erase what is obviously nonsensical to prevent such an impression?

2/ Frankly, in paragraph 3.2.2 (and elsewhere), I was unable to recognize the profit from mixing disparate disease data, which the authors endeavour to explain in their letter. My insufficient insight into intricacies of multiway correspondence analysis may be responsible. Hope, readers of IJERPH will perform better… In any case, for those less gifted the authors may add an uncomplicated explanation of e.g. how the “effective use” of the controls/disease combinations manifests in Fig. 5.

3/ From my experience, tick-borne disease data is typically overdispersed, and the Nordic data is by no means exceptional. A strong extra-Poisson component of variance can be attributed to both extrinsic (e.g. climate fluctuations) and intrinsic (e.g. feedback cycling in factors that reciprocally interplay such as pathogen’s prevalence and herd immunity in hosts) factors in the disease system. For instance, up to five periodicities (depending on region) can be distinguished in TBE data explaining almost 2/3 of total variance. For such a kind of data it is difficult to find a suitable statistical model of variance and bootstrapped confidence interval is appropriate. As to its questioned reliability, if I used block resampling in the simulations (that better preserves existing patterns in the data compared to the ‘white-noise’ approach), much wider confidence intervals resulted. I deliberately opted for the ‘white-noise’ model in order not to apply too strict criteria – quite in line with the authors’ arguments.  In this respect, the bootstrapped variance was kept rather moderate than “being quite large”.                                                        Apart from potential ‘decrease’, only two kinds of trend seem to me sound: WI (or simply ‘increase’), and P, and all districts should have been classified into these 2-3 categories. The other suggested trend categories (ID, IP, DP) most likely do not have any causal connection with the environmental changes in question – their prototypal patterns demonstrated in Fig.1 can easily be explained in terms of superposition of autonomous disease cycles. By the way, I cannot see much difference between the P- and DP-incidence series shown in Fig.1 (except for an anomalous initial upsurge in P). The authors should select better examples that would clearly illustrate the difference.

P.1, l.2: insert “Lyme” before “borreliosis”, pls. - I recommend to use consistently the term “Lyme borreliosis” instead of sole “borreliosis” throughout the article in order to distinguish from borreliae of the recurrent fever group co-circulating in the same ticks.

P.2, l.42-3: delete “mainly” before “tick-borne”, pls.

P.2, l.43:  …Lyme borreliosis, a bacterial disease caused by Borrelia burgedorfi sensu lato (BOR), and…

P.2, l.47-8: pls., delete the sentence – “As these diseases … discuss tularemia.” – it is not so! Anaplasmosis (i.e. that caused by A.phagocytophilum) and babesiosis are obligatorily transmitted by ticks, whereas tularemia is only facultatively tick-borne..

P.2, l.53: …ticks of the genera…

P.2, l.54: substitute “tick-borne disease agents” for “tick-borne diseases”, pls.

P.2, l.55: substitute “the agents in nature” for “and/or … the disease”, pls.

P.3, l.84: redefinition of “BOR” and “TBE” – delete it, pls.

P.3, l.118: Ixodes spp.

P.3, l.119: ..feed on small mammals..

P.4, l.128: substitute “preferred as hosts by” for the 2nd “for”, pls.

P.4, l.129: substitute “mammals preferred by” for “cervids … for”, pls.

P.4, l.138-9: ..ticks carrying borreliae are more widely spread than the TBE virus-infected ones which are more concentrated..

P.4, l.154-5: ..For Lyme borreliosis..

P.4, l.159-61: ..For TBE, wood and field mice, bank voles, and shrews are competent hosts [50]..

P.4, l.163: Ixodes spp.

P.4, box l.14: “Lyme disease“, pls.! It is not named after a person, so not “Lyme’s”..

P.4, box l.20-1: pls., delete “as well as partly … tick species vectors” – it could be applied to global distribution, not Fennoscandia…

P.4, box l.24-5: pls, delete “in which some domestic animals might be included” – it is not so!

P.4, box l.28: delete “the” before “I. persulcatus”, pls.

P.4, box l.29-30: Humans may also get infected by consuming unpasteurized milk products from goats…

P.6, l.182: “56%” should read “57%”

P.21, l.630: insert “Murmansk” and  “Karelia”, pls.

P.22, Table A2: the authors use the same abbreviation, “TBE”, for “Tree Broadleaf Evergreen” as well as tick-borne encephalitis (!!) – revise it, pls.

P.23, l.699: indicates

P.24, l.718: insert “References”, the section title, pls.

Author Response

see attached pdf (Letter_To_Editors_R2_AND_ResponseToReviewers#1.pdf)

Round 3

Reviewer 1 Report

Leibovici et al.:  Associating land cover changes with patterns of incidences of climate sensitive infections: an example on tick-borne diseases in the Nordic area. / 2nd revision

I acknowledge another partial amendment of the manuscript. I no longer comment on the authors’ disregard for some well-intended suggestions - There are three sorts of people: those who see, those who see when they are shown, and those who do not see (Leonardo da Vinci)… I cannot, however, overlook their adherence to obvious fallacies.

Concerning the conjectural reservoir role of domestic animals (p.4, text box): Once in my career I came across a case of TBE infection in laboratory personnel passaging the virus in mice. Since the laboratory mouse is counted among domestic animals, in terms of formal logic it indeed means that domestic animals may play a role of reservoir hosts… But this is the only case I know! There is so far no evidence that any other domesticated species, inclusive of dairy animals, can sustain perpetuation of the virus. And, by definition, the reservoir host is such a host that - along with the vector - keeps the agent in existence; any non-reservoir host brings it to an extinction. The authors should read the sources they are quoting carefully: [51] deals with seroprevalence, and whatsoever high titre in an animal means only that it has been challenged with the agent, not it is its reservoir. People living in endemic areas are also quite often TBEV-seropositive – does in mean they are reservoirs? [50] is a good review of TBEV reservoirs comprising, nevertheless, a speculative note that “..Dermacentor reticulatus ticks surpass the number of Ixodid tick bites on large domestic and game animals, leading to a potential additional circulation cycle of TBEV..” (besides, a lapse happened to authors of this claim in that D. reticulatus itsef is classified among Ixodid ticks). Reasons for declining it as a pure speculation predominate, e.g.: “It is unlikely that D. reticulatus can maintain TBEV in enzootic cycles in Europe as there is only a very short interval for the possibility of co-feeding larvae and nymphs” (Parasites & Vectors 9:314).

P.2, l.48: “biobiosis” – a new tick-borne disease?

Author Response

#1

I acknowledge another partial amendment of the manuscript. I no longer comment on the authors’ disregard for some well-intended suggestions - There are three sorts of people: those who see, those who see when they are shown, and those who do not see (Leonardo da Vinci)… I cannot, however, overlook their adherence to obvious fallacies.

Reply:

We are very sorry to disappoint you. I don’ t think we disregarded any of your comments or suggestions but it is true we didn’t necessarily agree. I like your quote from L. da Vinci though.

#1

Concerning the conjectural reservoir role of domestic animals (p.4, text box): Once in my career I came across a case of TBE infection in laboratory personnel passaging the virus in mice. Since the laboratory mouse is counted among domestic animals, in terms of formal logic it indeed means that domestic animals may play a role of reservoir hosts… But this is the only case I know! There is so far no evidence that any other domesticated species, inclusive of dairy animals, can sustain perpetuation of the virus. And, by definition, the reservoir host is such a host that - along with the vector - keeps the agent in existence; any non-reservoir host brings it to an extinction. The authors should read the sources they are quoting carefully: [51] deals with seroprevalence, and whatsoever high titre in an animal means only that it has been challenged with the agent, not it is its reservoir. People living in endemic areas are also quite often TBEV-seropositive – does in mean they are reservoirs? [50] is a good review of TBEV reservoirs comprising, nevertheless, a speculative note that “..Dermacentor reticulatus ticks surpass the number of Ixodid tick bites on large domestic and game animals, leading to a potential additional circulation cycle of TBEV..” (besides, a lapse happened to authors of this claim in that D. reticulatus itsef is classified among Ixodid ticks). Reasons for declining it as a pure speculation predominate, e.g.: “It is unlikely that D. reticulatus can maintain TBEV in enzootic cycles in Europe as there is only a very short interval for the possibility of co-feeding larvae and nymphs” (Parasites & Vectors 9:314).

Reply:

We agree that the evidence is not very strong as we had suggested by "might be”, the sentinel aspect as you mentioned is still appropriate. We’ve decided to remove this part of sentence as not essential for the paper.The new sentence(s) can read:

“Tick-borne encephalitis (TBE) is caused by a flavivirus that is mainly transmitted to humans by ticks infested when taking blood meals from competent vertebrate reservoir hosts, mainly small mammals [50]. …. Humans may also get infected by consuming unpasteurised milk products from goats, sheep and cows, which is a recurrent concern in areas with established tick populations and dairy production from domestic animals considered to be sentinel hosts for TBE virus (TBEV) [52,53]. ”

So, removing the “domestic” aspect in the first sentence concerning reservoir hosts and completing 

the last sentence about dairy production with “”from domestic animals considered to be sentinel hosts for TBE virus (TBEV) [52,53].

(Note, this is changing [refs numbers], the previous [51], Springer et al.  is now [53]. [52] being already cited here (Wallenhammar et al. )

#1

P.2, l.48: “biobiosis” – a new tick-borne disease?

Reply: 

Thanks for noticing the typo.